# Can Exercise Affect the Pain Characteristics in Patients with Fibromyalgia? A Randomized Controlled Trial

**DOI:** 10.3390/healthcare10122426

**Published:** 2022-11-30

**Authors:** Sotiria Vrouva, Varvara Sopidou, Evangelia Koutsioumpa, Konstantinos Chanopoulos, Alexandra Nikolopoulou, Vasileios Papatsimpas, George A. Koumantakis

**Affiliations:** 1Department of Physical Therapy, 401 Army General Hospital of Athens, 11525 Athens, Greece; 2Physiotherapy Department, School of Health and Care Sciences, University of West Attica (UNIWA), 12243 Athens, Greece; 3Laboratory of Neuromuscular and Cardiovascular Study of Motion (LANECASM), Physiotherapy Department, School of Health and Care Sciences, University of West Attica (UNIWA), 12243 Athens, Greece; 4Department of Biomedical Sciences, School of Health and Care Sciences, University of West Attica (UNIWA), 12243 Athens, Greece; 5Department of Intensive Care Unit, Larissa University General Hospital, 41334 Larissa, Greece; 6Computational Mathematics and Decision Making, 401 Army General Hospital of Athens, 11525 Athens, Greece; 7Laboratory of Advanced Physiotherapy (LAdPhys), Physiotherapy Department, School of Health and Care Sciences, University of West Attica (UNIWA), 12243 Athens, Greece

**Keywords:** fibromyalgia, active exercise, deep breath, chronic pain, pain characteristics

## Abstract

Exercise is often recommended for fibromyalgia. The aim of this study was to investigate the possible influence and change in the pain characteristics of fibromyalgia patients when breathing exercises were added to their exercise program. A total of 106 patients were included and randomly divided into two groups. Τhe first group of patients followed a program of active exercises up to the limits of pain, lasting 30 min with a repetition of two times a week. Patients of the second group followed the same program with the addition of diaphragmatic breaths when they reached the pain limit. The patients completed three questionnaires: the Fibromyalgia Rapid Screening Tool (FiRST), the Brief Pain Inventory (BPI), and the Pain Quality Assessment Scale (PQAS)—once at the beginning, once again after three weeks of exercise, and again 3 months since the beginning of the program. Independent *t*-tests for the mean total change scores in pain scales demonstrated that for the second group there was a greater improvement in all pain scales, except for the PQAS Deep Pain subscale (*p* = 0.38). In conclusion, both groups showed significant improvement in all characteristics of the pain scales; however, the improvement of the second group was significantly higher.

## 1. Introduction

Fibromyalgia is a complex syndrome, with its etiology still under debate [1]. Sufferers complain of fatigue and diffuse pain, muscle tenderness, stiffness, and insomnia [2]. It is the third most frequent musculoskeletal condition, and whilst still being relatively unknown, its frequency rises with age [3]. It mainly concerns women of the adult population [2]. The exact cause of fibromyalgia is unknown, but current theories suggest that a combination of factors, including genetic predisposition, stressful life situations, and peripheral (inflammatory) and central (cognitive–emotional) processes interact to cause altered pain perceptions due to neuromorphological changes (also known as “nociplastic pain”) [3]. Several studies have targeted factors related to the genetic expression of the genes responsible for kinase production [1]. Environmental factors, such as diet, obesity, and celiac disease, have also been implicated [1]. The assessment of patients includes their neurocognitive, functional, and psychological status [1]. The diagnosis of the disease is based on a physical examination with the defined criteria [1]. Comorbidity often prevents early recognition of the condition [1]. Numerous terms are used by fibromyalgia patients to describe their pain, which can impact the entire body and is also one of its main features [3].

Various pharmacological or non-pharmacological treatments are recommended for the treatment of fibromyalgia [1]. Fibromyalgia management calls for a multifaceted approach that incorporates patient education, behavioral therapy, exercise, and pain management [2,3]. Medication administration focuses primarily on managing chronic pain and treating mood [4]. Exercise is frequently suggested to enhance patients’ physical condition and quality of life, whether or not they are receiving medicine [4,5]. The focus is mainly on aerobic and flexibility exercises [4]. Flexibility exercises lead to a progressive increase in range, a reduction in pain, and the feeling of fatigue [4]. The combination of these exercises proves to be extremely effective [5]. At the same time, treatments that include exercise and breath management, such as yoga, appear to be effective, in addition to reducing pain and improving sleep and quality of life [5]. The results of these methods last longer than other types of exercise [5].

Among the latest revised recommendations for fibromyalgia management by EULAR, the type of exercise was among the key research priorities [5]. Therefore, the aim of this study was to investigate the extent of the influence in the pain characteristics of patients with fibromyalgia if breathing exercises were added to their exercise program.

## 2. Materials and Methods

This was a double-blind (participants and assessor) randomized controlled trial. The sample included 143 outpatients with fibromyalgia, and physical therapy was recommended by the orthopedic referral. There were 112 patients in all who received trial information, and six of them were excluded (four did not meet the inclusion criteria, and two did not wish to participate); therefore, 106 patients were finally included in the trial. Equal 1:1 randomization was utilized for allocation in either of the two groups based on the patient order of entry to the study. An independent research assistant was involved in the allocation process. The treatment interventions were carried out by the same physiotherapist. All patient examination data were documented by a physiotherapist who was blind to the study and independent.

All patients were recruited from the orthopedic clinics of the 401 General Military Hospital of Athens. All patients provided written informed consent prior to their participation to the study. The study protocol received approval from the bioethics committee of the hospital and followed the principles of the Declaration of Helsinki. The study excluded patients with open wounds, pregnant women, those with any form of neoplastic disease, those with respiratory, metabolic, or rheumatic diseases, as well as those with pacemakers or major cardiovascular diseases (Figure 1).

### 2.1. Procedure

The pain characteristics of included patients were recorded through the completion of 3 questionnaires: once before starting the exercise, once again after a three-week period of exercise, and again 3 months later after the initiation of the program.

The Fibromyalgia Rapid Screening Tool (FiRST) is a self-report scale used to identify fibromyalgia syndrome in people with diffuse persistent pain. The highest rate of accurate patient identification was achieved with a cut-off score of 5, which corresponds to the number of affirmative items [6,7].

The Brief Pain Inventory (BPI) is a self-report tool that has evolved into a standard for evaluating pain and its effects. The seven interference items are presented on a scale of 0 to 10, with 0 denoting no interference and 10 denoting complete interference. A recent consensus panel advised that the two BPI domains—pain intensity (severity) and the interference of pain with functioning—should be included as assessment criteria in all therapeutic studies for chronic pain [8]. If more than 50%—or four out of the seven interference items—have been filled on a given administration, their mean score may be used.

The Pain Quality Assessment Scale (PQAS) is a 20-item scale, with individual items scored between 0 and 10, except for the last item that is scored on a three-level categorical scale. It is not advised to compute a “global” PQAS scale score, as this would likely omit crucial information about certain aspects of pain. Each item can be separately rated to better understand the types of pain the patients feel. Item 20 describes the temporal pain characteristics and is scored categorically (three options available on three types of temporal patterns that respondents might indicate). The mean score of the items related to each subscale (Paroxysmal, Surface, and Deep) is usually separately presented. Only respondents who have responded to every item should have their sub-scale scores calculated [9]. No item score is to be extrapolated from other items for any missing items; they are classified as missing data.

All questionnaires were translated and validated in the Greek language [6,8], except PQAS, which the present study served as a means for translation and validation in Greek.

### 2.2. Interventions

The same 10-min warm-up routine was applied to both groups, consisting of active mobilization and 30-second-long stretches for the spine, hips, knees, ankles, shoulders, elbows, and wrists [4,10,11]. The first group of patients followed a program of dynamic exercises for the deltoids, quadriceps, trunk extensions, hip extensions, elbow flexors, and gastrocnemius that lasted 30 min and were administered twice a week [4]. The patients performed a set of 10 repetitions for each muscle group [4], with a break, as per their convenience, at the pain limit.

Participants of the second group followed the program of the first group exactly, and diaphragmatic slow breaths were additionally performed, where the pain limit was reached each time. Patients performed the first exercise, discovering what the end of their range was (the point when the movement became painful) [9]. At this point, they were instructed to take three breaths in a slow manner and then to initiate and complete the program as described in the first group, though whilst performing deep breaths upon the end of the range of each exercise.

### 2.3. Statistical Analysis

An a priori power analysis was performed, aiming at estimating the patient sample size required for results to have adequate power for the between-group comparisons. The parameters of the power analysis included were: (a) an alpha level of 0.05 for the appropriate statistical tests for comparative analysis; and (b) a statistical power of at least 80%. (c) The effect size for the evaluation of the change between time points (baseline, 3 weeks, and 3 months) was assigned to be small (f = 0.23) due to the short duration of the interventions, and medium (f = 0.55) for the difference between two independent means (depending on the program they followed) in order to reduce the probability differences due to chance. The required number of participants was calculated using the software G*Power 3.0.10 by entering the above-mentioned parameters. The calculator indicated that by using 53 patients in each group, a statistical power level of more than 80% could be achieved for all assessments performed.

The effect of both exercise programs (exercise alone, and exercise combined with breathing) on clinical pain characteristics (BPI Severity and BPI Interference pain subscales, the FiRST, and the PQAS subscales: Paroxysmal, Surface and Deep pain) was examined with a mixed-model ANOVA analysis, with repeated measures for the within-subjects time factor (at baseline, 3 weeks later, and after 3 months) and a between-subjects factor between the two different groups.

Independent t-tests were initially carried out for all the mentioned variables to assess whether the two groups were comparable at baseline. Furthermore, in order to investigate the differences in total improvement between the two programs of exercise after the completion of the intervention, independent t-tests were performed at the end of the intervention for each pain scale.

Possible dependencies between qualitative variables and categorical pain characteristics of the two different exercise groups were investigated by the use of Pearson’s chi-squared test (χ^2^).

Categorical characteristics of patients are presented as a number of patients (n) or percentages (%), while data of descriptive statistics are presented as mean ± standard deviation (SD). Statistical analysis was performed using IBM SPSS Statistics v.28 for Windows (SPSS Inc., Chicago, IL, USA), and statistical significance for all tests was set at 0.05. 

## 3. Results

The study involved a total of 106 patients with chronic pain, diagnosed with fibromyalgia, aged 35 to 57 years (46.83 ± 5.991). Of these patients, 45.3% (n = 48) were male and 54.7% (n = 58) were female, where about half of them were receiving some kind of treatment or medication. The majority of patients 72.7% (n = 77) indicated that they felt pain in the back area. Similarly, 77.4% (n = 82) described their pain as deep, as opposed to 22.6% (n = 24) who rated it as superficial. Regarding the different time qualities of pain, 66% (n = 70) had stable pain, 23.6% (n = 25) had variable pain, and only 10.4% (n = 11) felt pain intermittently. In addition, the FiRST self-questionnaire score for the detection of FM showed that about 60% (n = 63) of the patients met the cut-off score ≥ 5/6 criterion. Mean and categorical scores for all patients in pain subscales are presented in Table 1.

The results of the independent t-tests and chi-squared tests revealed that there were no statistically significant between-group differences before the implementation of the exercise programs. Table 2 shows no differences in mean values or in initial frequencies between the two groups, as well as the *p*-values for the respective tests.

The mixed-model ANOVA results within the three time-points showed statistically significant differences for both groups between the mean values of all variables examined (*p* < 0.05). Table 3 shows the mean scores in the pain scales for every time-point for each group, and the results of the mixed ANOVA for the within-subjects’ factor (time).

The results of multiple comparisons per pair of assessment time-points (Bonferroni post hoc tests) showed that this significant improvement appears to take place immediately after the three-week intervention and is of equal amount for both of the two groups and remains constant until the last assessment three months from baseline (as attested by the non-significant interaction effect). The results of the between-group comparisons (between-subjects factor) performed for the two different exercise programs showed statistically significant differences in mean scores for BPI subscales (Severity F(1, 104) = 37.8, *p* < 0.001 and Interference F(1, 104) = 29.42, *p* < 0.001), the FiRST scale F(1, 104) = 37.14, *p* < 0.001 and for two out of the three PQAS subscales (Paroxysmal F(1, 104) = 8.43, *p* = 0.005 and Surface F(1, 104) = 13.08, *p* < 0.001), but the difference between the two groups for the PQAS subscale of Deep Pain was not statistically significant (F(1, 104) = 0.59, *p* < 0.446).

Furthermore, the results from independent t-tests for the mean total change improvement in each pain scale showed that the second group registered a greater change improvement in all pain scales. These differences were statistically significant (*p* ≤ 0.001) for all pain scales, except for the PQAS Deep Pain subscale (*p* = 0.38), as depicted in Figure 2 below.

## 4. Discussion

According to recommendations for the effective management of fibromyalgia, exercise has a prominent and advantageous role [5]. Therefore, two different exercise programs were tested to establish the added benefit of deep breathing exercises to a dynamic exercise program. Previous research has demonstrated that low levels of flexibility have been associated with postural problems, pain, injuries, reduced local vascularity and increased neuromuscular tensions. For this reason, the use of warm-ups and stretches has been deemed necessary [4]. However, flexibility levels are of no clinical significance if they cannot provide guaranteed pain reduction. Couto et al. [12] stated that rehabilitation programs using stretching can reduce pain intensity, while others have reported that it is almost certain that flexibility cannot contribute to decreased pain [4].

Compared to other types of exercise, aerobic exercise has been preferred [12]. Aerobic exercise has been documented in many studies as the appropriate exercise to reduce pain levels [13]. Mixed-type exercise combining aerobic exercise with strength training is unclear in its effectiveness on pain in patients with fibromyalgia [14]. In contrast, several studies have noted that high-intensity exercise could lead to increases in pain as it blocks the anti-inflammatory response [15]. Conversely, long-term aerobic exercise does not show satisfactory results [16]. The present study cannot agree or disagree with this, as the measurements were not repeated after 3 months.

In our study, aerobic exercise appeared to have a beneficial impact in both groups, since we found statistically significant differences in all scales, between baseline and after the completion, for each group. In people suffering from fibromyalgia, it is often observed that complete rest, as well as a small daily amount of movement, can lower painful sensations [17]. Induction of this program more than twice per week is not recommended, as it has been proven to lead to increased fatigue and discomfort [17]. Our study agrees on exercise frequency with many previous studies [16]. It is generally stated that the duration of the exercise should not exceed half an hour, with a frequency of about two times a week [16].

Both groups showed significant improvement in all characteristics of the pain scales, but the improvement of the second group was significantly higher. The recommendation of the aerobic exercise and breathing combination is not only to increase fitness, but also to provide a way of controlling allodynia and hyperalgesia, which we are aware exists in patients with fibromyalgia. Controlling the pain limit with breathing seems to have an effect because patients who suffer from central pain, such as fibromyalgia [18], learn through this technique that reaching the point of pain is safe [19,20,21]. Deep slow breathing techniques decisively affect autonomic processing and pain thresholds [22]. For this reason, deep slow abdominal breathing with instructions to inhale through the nose and exhale through the mouth can lead to the control of, and further reduction of, pain [23].

People with chronic pain have a wrong perception of the force they have to exert for exercise performance, resulting in increased sensory feedback [24]. The recommended rate is 4 s for the inspiratory, and 5 s for the expiratory phase of breathing, respectively [25], and repetitions at a slow rate [26].

Any potential threat to the body can increase pain [20]. Thus, a revision of the pain threshold is needed. By controlling breathing at the limits of the pain-inducing trajectory, the brain essentially reassures itself that the body is not in danger [20]. Adding guided movement that is performed consistently can be vital to induce desired behavior changes [21].

## 5. Conclusions

A significant difference in pain reduction appeared to emerge after the three-week intervention for both groups, and remained significant until the last assessment, three months from baseline. Both groups showed significant improvement in all characteristics of the pain, as measured by three relevant pain scales, but the improvement of the second group was significantly higher. It is worth mentioning that this greater improvement in the pain characteristics of the second compared to the first group occurred in a short period of only three weeks.

## Figures and Tables

**Figure 1 healthcare-10-02426-f001:**
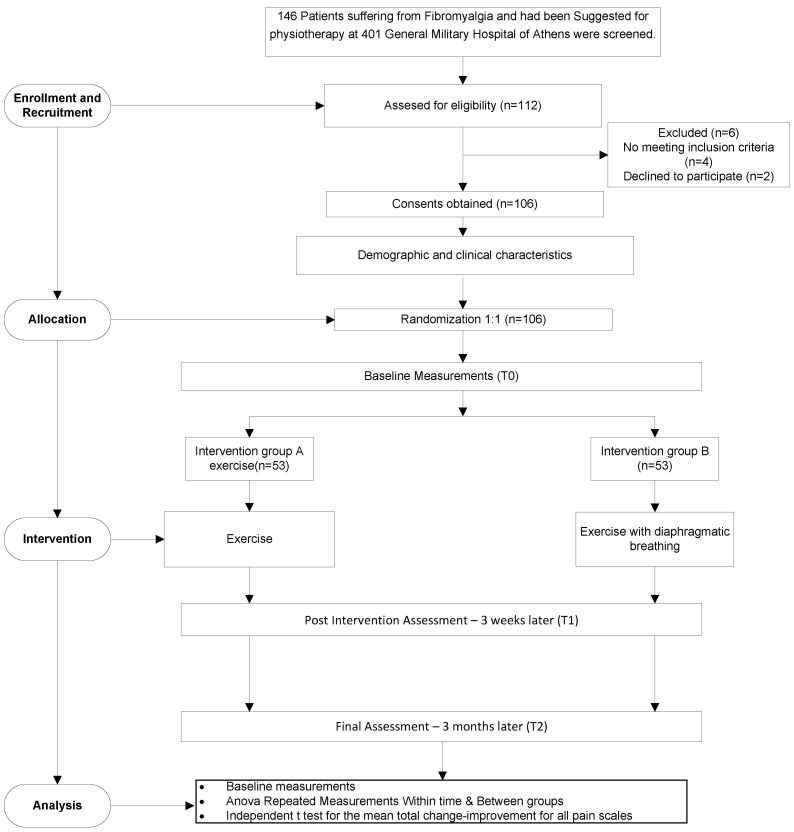
Flow diagram of patient progress through the trial.

**Figure 2 healthcare-10-02426-f002:**
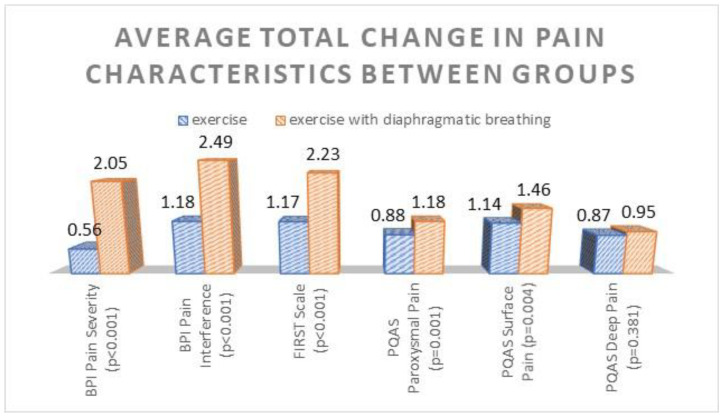
Average total change improvement in pain scales, estimated on marginal means for each group. *p*-values < 0.05 indicate statistically significant differences between groups.

**Table 1 healthcare-10-02426-t001:** Demographic data and pain characteristics of the subjects ^a^ (n = 106).

Age	46.83 ± 5.991	(35–57)
Sex ^b^ (male/female)	48 (45.3)/58 (54.7)	
Areas feeling Pain ^b^ (Axial region/back/back & leg/shoulder)	17 (16)/71 (67)/6 (5.7)/12 (11.3)	
Treatment or medication ^b^ (yes/no)	58 (54.7)/48 (45.3)	
FiRST total score ^b^ (<5/≥5)	43 (40.6)/63 (59.4)	
BPI Pain Severity Score	6.09 ± 0.769	(4.5–7.75)
BPI Pain Interference Score	8.06 ± 0.812	(6–10)
PQAS Paroxysmal pain	6.08 ± 0.491	(4.6–7.2)
PQAS Surface pain	6.30 ± 0.582	(4.8–7.4)
PQAS Deep pain	5.71 ± 0.507	(4–7.2)
PQAS (deep/superficial) pain ^b^	82 (77.4)/24 (22.6)	
PQAS different time qualities of Pain ^b^ (intermittent/stable/variable)	11 (10.4)/70 (66)/25 (23.6)	

^a^ Values are expressed as mean ± SD (range); ^b^ values are expressed as number of patients (%).

**Table 2 healthcare-10-02426-t002:** Baseline demographic and pain characteristics between groups.

	Exercise (n = 53)	Exercise with Diaphragmatic Breathing (n = 53)	*p*-Value
Age	47.04 ± 6.28	46.62 ± 5.73	0.72
Sex ^b^ (male/female)	21 (39.6)/32 (60.4)	27 (50.9)/26 (49.1)	0.24
Treatment or medication ^b^ (y/n)	34 (64.2)/19 (35.8)	24 (45.3)/29 (54.7)	0.05
BPI Pain Severity Score	5.99 ± 0.83	6.18 ± 0.69	0.21
BPI Pain Interference Score	7.98 ± 0.90	8.13 ± 0.72	0.34
PQAS Paroxysmal pain	6.08 ± 0.50	6.07 ± 0.48	0.97
PQAS Surface pain	6.31 ± 0.57	6.28 ± 0.60	0.79
PQAS Deep pain	5.71 ± 0.55	5.70 ± 0.47	0.97
PQAS (deep/superficial) pain ^b^	42 (79.2)/11 (20.8)	40 (75.5)/13 (24.5)	0.64
PQAS diff. time qualities of pain ^b^ (intermittent/stable/variable)	6 (11.3)/34 (64.2)/13 (24.5)	5 (9.4)/36 (67.9)/12 (22.6)	0.91

^b^ values are expressed as the number of patients (%).

**Table 3 healthcare-10-02426-t003:** Mean scores in the pain scales for every time-point for each group, and results of the mixed ANOVA for the time factor (F-within and *p*-value).

Pain Scale	Group	Baseline	3 Weeks	3 Months	F	*p*-Value
BPI Severity	Exercise	6.00 ± 0.11	5.48 ± 0.10	5.43 ± 0.10	373.6	<0.001
	Exercise + diaphragmatic breathing	6.18 ± 0.11	4.15 ± 0.10	4.13 ± 0.10		
BPI Interference	Exercise	7.98 ± 0.11	6.85 ± 0.10	6.80 ± 0.10	853.7	<0.001
	Exercise + diaphragmatic breathing	8.13 ± 0.11	5.63 ± 0.10	5.64 ± 0.10		
FiRST	Exercise	4.77 ± 0.12	3.87 ± 0.14	3.60 ± 0.14	240,52	<0.001
	Exercise with diaphragmatic breathing	4.55 ± 0.11	2.43 ± 0.10	2.32 ± 0.10		
PQAS Paroxysmal	Exercise	6.08 ± 0.07	5.28 ± 0.06	5.21 ± 0.06	498.76	<0.001
	Exercise with diaphragmatic breathing	6.08 ± 0.07	4.93 ± 0.06	4.90 ± 0.06		
PQAS Surface	Exercise	6.32 ± 0.08	5.23 ± 0.05	5.18 ± 0.05	536.49	<0.001
	Exercise with diaphragmatic breathing	6.29 ± 0.08	4.83 ± 0.05	4.82 ± 0.05		
PQAS Deep	Exercise	5.71 ± 0.07	4.89 ± 0.08	4.84 ± 0.07	342.37	<0.001
	Exercise with diaphragmatic breathing	5.71 ± 0.07	4.77 ± 0.08	4.75 ± 0.07		

## Data Availability

The data presented in this study are available on request from the corresponding author. The data are not publicly available due to privacy restrictions.

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
