# Peer review of "Can Exercise Affect the Pain Characteristics in Patients with Fibromyalgia? A Randomized Controlled Trial"

_healthcare, 2022, doi:10.3390/healthcare10122426_

Round 1
Reviewer 1 Report
The manuscript is well written and the study is well designed. The discussion is adequate and the authors have placed the study well in the scientific field.
I have only one concern. I can't see the statistical significance in the figures or even in the tables. The text says that there is a significant improvement in pain scores in the group that used diaphragmatic breathing during the exercises, but I could not find any indication of significance.
Also, the authors used the t-test for statistical analysis, which is not correct here. A repeated measures ANOVA is the correct statistic for the calculation. I strongly recommend recalculating the results using the repeated measures ANOVA and indicate the significance in the figures.
Author Response
"Please see the attachment."

Reviewer 2 Report
Dear Authors, this is an interesting paper regarding the issue of pain in fibromyalgia. It is well-written and scientifically sound, only I would suggest:
1) to include slightly more information in the introduction, e.g. regarding its measurements
2) I am not sure about figure 4, whether it is necessary/clear.
Kind regards
Author Response
"Please see the attachment."

Reviewer 3 Report
The authors have reported the effect of breathing exercises on the pain characteristics in fibromyalgia, the results agreeing with the emerging body of working in recent areas on the benefits of breathing exercises on pain management in fibromyalgia.
Overall the study is well designed and manuscript is well-written, the experimental design clearly explained and the results are very convincing. The results appears to agree with other similar studies in this research area.
Just a few comments:
1) Are any of the results statistically significant? Is it possible to add in error bars or show statistical significance on the graphs
2) Does Tables 3 contain the same informations as the graphs, and if so, is it necessary?
3) Figures 2, 3 and 4: presentation of the results e.g. the grey background against blue and green line makes the data somehow "buried" into the background. Also, the graphs as pictures have a "fuzziness" about them, making the axis label, figure legend etc really hard to read. Please ensure the diagrams are of suitable resolution when uploading.
4) Figure 5: some presentation issues. The x-axis appeared lopsided and the values on top of each bar appeared to overlap.
5) The pain scores appear to plateau between 3 weeks and 3 months for both types of exercise. Would other means of intervention in supplement to these exercises improve the out come? any comments
6) More speculation: for the group doing the normal exercise, would switching over to the breathing exercise at the end of the three months improve their pain score?
Author Response
"Please see the attachment."

Reviewer 4 Report
The result in the paper is clear; that group 2 (exercise + breathing) shows significantly better effect than group 1 . However, I have a major concern about the methodology. The group 1 patients perform a set of 10 reps of each exercise without a break in between, while group 2 patients can take a break when they feel heavy pain. So, may be this effect we see is just a consequence of taking a break to relax and spread out the exercise time, and it has nothing to do with any specific breathing regimen they follow during the breaks.
I think a third group of patients is needed to tease out this possibility, wherein the patients take a break within the 10 reps (as per their convenience) without any specific breathing instructions. Otherwise you cannot claim any significant effect due to introduction of breathing routines
In line 241 ( conclusion section), I think the authors have mixed up group 1 and group 2.
Author Response
"Please see the attachment."

Round 2
Reviewer 1 Report
The authors have addressed all my concerns. The manuscript can be published in present form.
Reviewer 4 Report
I'm satisfied with the author's response